# Appendiceal microbiome in uncomplicated and complicated acute appendicitis: A prospective cohort study

Sanja Vanhatalo[1,2], Eveliina Munukka[3,4], Teemu Kallonen[1,2,3], Suvi Sippola[5,6], Juha Grönroos[5,6], Jussi Haijanen[5,6], Antti J. Hakanen[1,2,3], Paulina Salminen[5,6]*

1 Research Center for Infections and Immunity, Institute of Biomedicine, University of Turku, Turku, Finland, 2 Laboratory Division, Department of Clinical Microbiology, Turku University Hospital, Turku, Finland, 3 Faculty of Medicine, Microbiome Biobank, University of Turku and Turku University Hospital, Turku, Finland, 4 Biocodex Nordics, Espoo, Finland, 5 Division of Digestive Surgery and Urology, Turku University Hospital, Turku, Finland, 6 Department of Surgery, University of Turku, Turku, Finland

* paulina.salminen@tyks.fi

**Data Availability Statement:** The clinical patient data is available only by request and this data includes sensitive information despite the deidentification and thus cannot be shared based

## Abstract

### Background

Uncomplicated and complicated acute appendicitis seem to be two different forms of this common abdominal emergency. The contribution of appendiceal microbiota to appendicitis pathogenesis has been suggested, but differences between uncomplicated and complicated appendicitis are largely unknown. We compared the appendiceal microbiota in uncomplicated and complicated acute appendicitis.

### Methods

This prospective single-center clinical cohort study was conducted as part of larger multi-center MAPPAC trial enrolling adult patients with computed tomography or clinically confirmed uncomplicated or complicated acute appendicitis. The microbial composition of the appendiceal lumen was determined using 16S rRNA gene amplicon sequencing.

### Results

Between April 11, 2017, and March 29, 2019, 118 samples (41 uncomplicated and 77 complicated appendicitis) were available. After adjusting for age, sex, and BMI, alpha diversity in complicated appendicitis was higher (Shannon p = 0.011, Chao1 p = 0.006) compared to uncomplicated appendicitis. Microbial compositions were different between uncomplicated and complicated appendicitis (Bray-Curtis distance, P = 0.002). Species poor appendiceal microbiota composition with specific predominant bacteria was present in some patients regardless of appendicitis severity.

### Conclusion

Uncomplicated and complicated acute appendicitis have different appendiceal microbiome profiles further supporting the disconnection between these two different forms of acute appendicitis.

on legal restrictions by Finnish GDPR rules. The point of contact is the Ethics Committee of Hospital District of Southwest Finland (eettinen. toimikunta@tyks.fi). The raw 16S-data set is available at the Sequence Read Archive (SRA) portal of NCBI under project number PRJNA869932 (http://www.ncbi.nlm.nih.gov/ bioproject/869932).

**Funding:** The MAPPAC study was supported by research grants from the Mary and Georg C. Ehrnrooth Foundation, the Sigrid Jusélius Foundation, the Finnish Academy, Government research grant awarded to Turku University Hospital (EVO foundation), The Maud Kuistila Memorial Foundation, Paulo Foundation, Doctoral Program in Clinical Research at the University of Turku, and Turku University foundation. The funders had no role in study design, data collection and analysis, decision to publish, or preparation of the manuscript.

**Competing interests:** EM is currently working as full-time Medical Advisor for Biocodex Nordics. PS reports receiving personal fees for lectures form Merck and Orion Pharma. AJH reports receiving personal fees for lectures from BioCodex, Merck and Pfizer. This does not alter our adherence to PLOS ONE policies on sharing data and materials. All other authors declare no competing interests.

## Study registration

ClinicalTrials.gov NCT03257423.

## Introduction

With current knowledge, acute appendicitis can be both epidemiologically and clinically classified by severity in two different entities of uncomplicated and complicated acute appendicitis [1–3] allowing a stratified approach to management [2]. The dogma of acute appendicitis inevitably always progressing to perforation without early appendicectomy seems to apply only to complicated acute appendicitis. As non-operative management of uncomplicated appendicitis has been identified as feasible and safe [4–9], it has become very relevant to clinically differentiate between uncomplicated and complicated acute appendicitis. This differential diagnosis still remains challenging and evident findings of complicated acute appendicitis include perforation, abscess or a suspicion of a tumor. The presence of an appendicolith has also been shown to be associated with a more complicated course of the disease [8,9] and should be considered a finding of complicated acute appendicitis not eligible for non-operative treatment [6,7,10]. Despite appendicitis being one of the most common general surgical emergencies worldwide, the ethiology still remains poorly understood and bridging the knowledge gaps in the pathogenesis of these different forms of appendicitis severity is necessary [2].

The understanding of both microbiological etiology of acute appendicitis and the potential differences in etiology and pathogenesis between uncomplicated and complicated acute appendicitis is very limited, but the involvement of bacteria or the appendicular microbiome has been suggested [2]. As majority of bacteria in the appendix are anaerobes, the role of culturing methods is limited and next generation sequencing (NGS) methods are somewhat a necessity in the detection of the microbiological factors in the appendicitis etiology. All studies that have characterised the appendiceal microbiome by NGS show that appendiceal microbiome is highly diverse harboring large interindividual variation in both healthy and inflamed appendixes [11–15]. Members of gram negative *Fusobacteria* were considered to be infectious, associating with more severe appendicitis in a study using fluorescence in situ hybridization (FISH) [16]. Studies using 16S rRNA gene amplicon sequencing with both adult and pediatric patients have indeed confirmed *Fusobacteria* as an important microbial factor [11,12,17–19]. In addition, increased levels of *Parvimonas* and *Porphyromonas* in pediatric appendicitis samples have been reported [17,18].

To our knowledge, only one previous study utilizing NGS has specifically compared the microbial composition of the appendix in adult patients with uncomplicated and complicated appendicitis showing no significant difference between perforated and unperforated acute appendicitis samples [14]. However, this study was limited by both a very small sample size and lack of standardized clinical definitions of appendicitis severity focusing on the methodological comparison of bacterial culture and NGS. In light of the limited available data, we performed this prospective study consisting of patients with CT-confirmed uncomplicated or complicated acute appendicitis with thorough prospective clinical data and standardized appendix samples aiming to assess the potential differences in appendiceal microbiota composition between uncomplicated and complicated appendicitis using 16S rRNA gene amplicon sequencing.

## Methods

### Study design and patients

This prospective cohort study was a single-center arm of a multicenter MAPPAC (Microbiology Appendicitis Acuta) trial (ClinicalTrials.gov NCT03257423) and was conducted at Turku

University Hospital in Finland. Study design and key methods have been reported previously [20]. The study enrolled adult patients with CT or clinically confirmed either uncomplicated or complicated acute appendicitis undergoing appendicectomy in order to have the appendix both as the reference standard for the clinical diagnosis and the availability of appendiceal samples. MAPPAC trial was conducted concurrently with two randomized controlled trials (RCTs) APPAC II [7] and APPACIII [10] enrolling patients with a CT-confirmed uncomplicated acute appendicitis. APPAC II was an open-label, noninferiority trial comparing oral moxifloxacin with intravenous ertapenem followed by oral levofloxacin and metronidazole. APPAC III was a double-blind, placebo-controlled, superiority study comparing antibiotic therapy (intravenous ertapenem followed by oral levofloxacin and metronidazole) with placebo in the treatment of uncomplicated appendicitis.

Intramural neutrophil invasion in the histopathological examination of the removed appendix was required for the diagnosis of acute appendicitis. Acute appendicitis was defined uncomplicated if no features of complicated acute appenditis was present. Complicated appendicitis was defined as the presence of an appendicolith, perforation, abscess, gangrene, or suspicion of tumor or the combination of these. To validate the accuracy of the differential diagnosis between uncomplicated and complicated acute appendicitis, all patients were assessed using clinical data, CT, surgical, and histopathological findings by two investigators (S.S. and J.H.) unaware of the other's evaluation. In cases of disagreement, the clinical diagnosis was reviewed by a third investigator (P.S.).

## Outcomes

The objective of this study was to compare the microbial composition of an inflamed appendix between uncomplicated and complicated acute appendicitis. Microbial composition was assessed by the relative levels of bacterial phyla, genera, and species. Species richness and diversity, as well as dissimilarity of microbiomes, i.e. beta diversity, were measured and differential abundance analysis was used to identify differentiating phyla, genera, and species associated with appendicitis severity.

## Sample collection and processing

Microbiological swabs from the appendiceal lumen were collected right after the appendicectomy. Sample collection and bacterial DNA extraction have been previously described in detail [20]. Briefly, extraction of DNA from appendiceal swabs was performed by semiautomated GXT Stool Extraction kit (Hain Lifescience GmbH, Nehren, Germany). DNA concentration was quantified with Qubit fluorometer (Life Technologies, Carlsbad, California, USA).

## 16S rRNA amplicon sequencing

16S rRNA amplicon sequencing was performed targeting the V3-V4 hypervariable region. Negative and positive control samples were included in the sequencing: negative DNA extraction control, negative PCR control, and a mock community (ZymoBiomics microbial community DNA standard, Zymo Research, Irvine, California, USA) as a positive control. Amplicon libraries were generated following the Illumina protocol (https://support.illumina.com/documents/documentation/chemistry_documentation/16s/16s-metagenomic-library-prep-guide-15044223-b.pdf) with the exception that increased amount of 75 ng of DNA template was used in the amplicon PCR reaction. Amplicon libraries were quantified using Qubit fluorometer and 10% of Phix (Illumina, San Diego, California, USA) was added to each equimolar pool. Sequencing was performed using MiSeq Reagent Kit v3 and paired-end 2×300 bp protocol on a MiSeq System (Illumina).

## Statistical and bioinformatical analysis

Group differences in numerical (age, body mass index (BMI)) and categorical (sex) baseline characteristics were tested with Wilcoxon rank sum test and Pearson´s chi-squared test, respectively. All bioinformatical analyses were performed in R 4.0.3. Amplicon sequence variants (ASVs) of the 16S rRNA gene sequences were formed using the dada2 package, version 1.18.0 [21]. Raw reads were quality filtered and trimmed using the function filterAndTrim. The ASVs were assigned to taxa using the SILVA rRNA (v. 138) database as reference [22–24]. Contaminants, eukaryotes and mitochondrial sequences were removed from bacterial data. Archaeal sequences remained in the data. Decontamination was performed using R package decontam with the method "Prevalence" and a threshold of 0.225 [25]. This method compares prevalence of each ASV in true samples to the prevalence in negative controls to identify contaminants. To ensure that decontamination worked well, results were inspected and manually corrected. In normalization and all further analyses, the samples with more than 1800 sequences were included. Shannon's index did not strongly correlate with library sizes of samples even in the in non-normalized raw data (Pearson correlation = 0.17). Thus, raw sequence counts of ASVs were transformed to relative abundances that were used in downstream analyses and for alpha diversity calculations [26]. Principal coordinate analysis (PCoA) was calculated using function cmdscale in R package vegan [27] and permutational multivariate analysis of variance (PERMANOVA), with control variables age, sex and BMI, was performed using function multiconstrained in R package BiodiversityR. Both analyses were based on Bray-Curtis dissimilarities. Significantly differentiating bacterial species, genus, and phyla between groups were detected using the DESeq2 method [28]. The p-values were adjusted using Benjamini-Hochberg method and adjusted p-values <0.01 were considered as statistically significant. All the reported p-values related to differential abundance analyses are adjusted p-values. Alpha diversity analyses were performed using standard linear regression models with alpha diversity measure (Chao1, Shannon entropy or number of observed species) as the response variable and appendicitis severity, age, sex, and BMI as the predictors.

## Study approval

This study has been accepted by the Ethical Committee of the Hospital district of Southwest Finland (Turku University Hospital) and Finnish Medicines Agency (Fimea). Trial was performed in accordance with the Declaration of Helsinki and all patients and in the case of minors their legal guardians gave written informed consent to participate in the study.

## Results

Altogether 308 patients were enrolled in the MAPPAC study between April 11, 2017, and March 29, 2019; Fig 1 shows the study flow. Out of the 169 patients with uncomplicated or complicated acute appendicitis undergoing appendicectomy, 118 (70%) samples were included in the final NGS analysis (41 uncomplicated, 77 complicated appendicitis) with 51 patients excluded due to failed library preparation (n = 26), incomplete sample collection (n = 21), or too low 16S amplicon library size (n = 4). Patient baseline characteristics and the subtypes of complicated appendicitis are presented in Table 1. The groups were similar regarding age and BMI, but the proportion of women in the uncomplicated appendicitis group was higher compared to the complicated appendicitis group (66% vs. 45%, respectively, p = 0.0346). Out of the 77 patients with complicated acute appendicitis, an appendicolith was present in 57 (74%) patients.

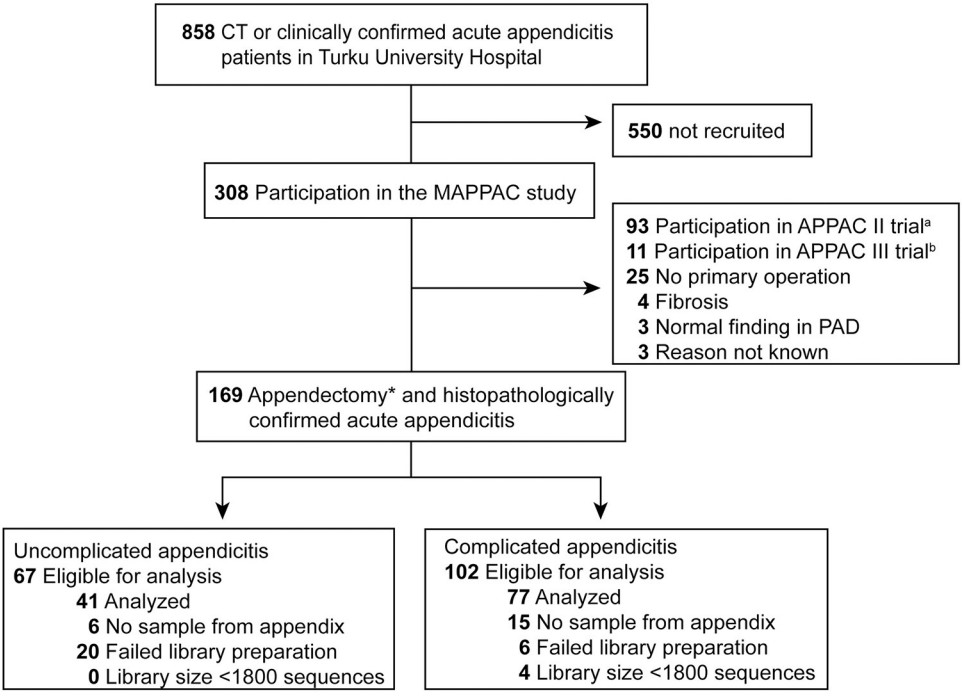

**Fig 1. Flow chart of the study.** *Appendectomy for primary acute appendicitis. Only patients treated with appendectomy and histopathologically confirmed acute appendicitis study were included in the analysis to have the microbiological sample and to have confirmation for the differential diagnosis. a) Randomized, multicenter, open-label, noninferiority clinical trial comparing oral moxifloxacin with intravenous ertapenem followed by oral levofloxacin and metronidazole. b) Randomized, multicentre, placebo-controlled, double-blind trial comparing antibiotic therapy with placebo in the treatment of uncomplicated acute appendicitis.

## Differing microbial signatures in uncomplicated and complicated acute appendicitis

Appendiceal microbiome composition in uncomplicated acute appendicitis was significantly different compared with complicated acute appendicitis regarding both alpha and beta

**Table 1. Patient baseline characteristics and subtypes of complicated appendicitis.**

| Characteristic | Uncomplicated appendicitis (N = 41) | Complicated appendicitis (N = 77) | P value |
|---|---|---|---|
| Sex, n (%) | | | |
| Female | 27 (66) | 35 (45) | 0.0346 |
| Male | 15 (34) | 42 (55) | |
| Age (years), median (range) | 37 (16–75) | 43 (18–69) | 0.7090 |
| Body mass index (kg/m$^2$), median (range) | 25.8 (20.4–39.5) | 26·6 (18·4–42.7) | 0.3395 |
| Complicated subtypes, n (%) | | | |
| Appendicolith | | 30 (39.0) | |
| Gangrenous/perforation | | 14 (18.2) | |
| Gangrenous/perforation and appendicolith | | 21 (27.3) | |
| Periappendicular abcess (circumscribed closed perforation) | | 6 (7.8) | |
| Gangrenous/perforation, abscess and appendicolith | | 3 (3.9) | |
| Abscess and appendicolith | | 1 (1.3) | |
| Tumour and appendicolith | | 1 (1.3) | |
| Gangrenous/perforation, tumour and appendicolith | | 1 (1.3) | |

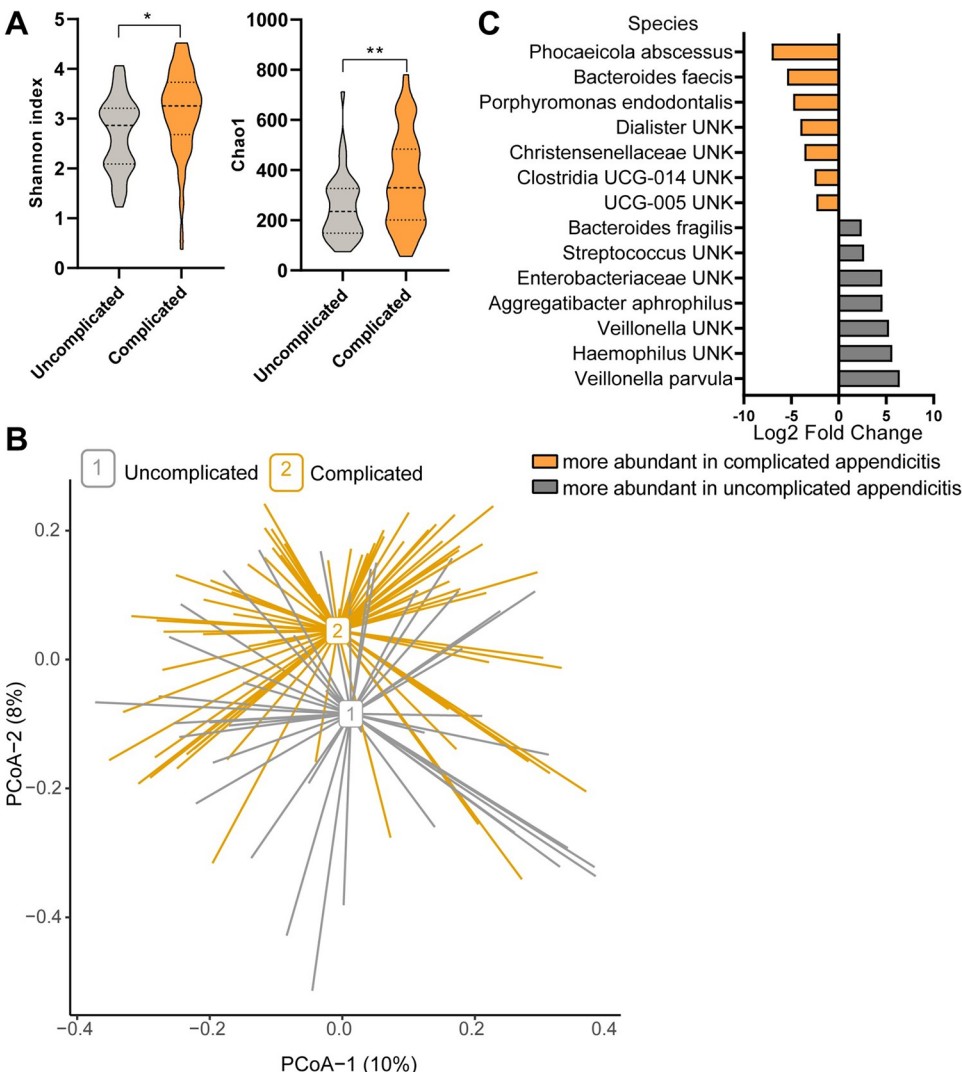

**Fig 2. The difference in appendiceal microbiota between uncomplicated and complicated acute appendicitis.** A) Violin plot representing alpha diversity measures Shannon index and Chao1 in uncomplicated and complicated acute appendicitis. * p < 0.05 and ** p < 0.01). B) Principal Coordinates Analysis (PCoA), i.e. beta diversity, based on Bray-Curtis distances. The percentage of variation explained by the two first PCoA dimensions is indicated on the respective axes. C) Barplot showing significantly (adj p < 0.01) differentiating species in uncomplicated compared to complicated that were present in meaningful levels (average relative abundance >0.1%). Species are listed along the y-axis and x-axis indicates the log2 fold change. UNK = unknown species.

diversities. Alpha diversity indices Shannon and Chao1 describing diversity and richness were lower in uncomplicated acute appendicitis compared to complicated appendicitis samples (Fig 2A, p = 0.011 and p = 0.006, respectively). Similar to alpha diversity, Principal coordinate analysis based on Bray-Curtis dissimilarity (Fig 2B) also indicated different beta diversities in appendiceal microbiota between the two forms of appendicitis severity. Pairwise comparisons revealed significantly different Bray-Curtis distance measure between the two groups (p = 0.002).

To investigate which bacteria are responsible for the observed beta diversity difference, the relative abundances of bacteria between groups were compared in phylum, genus, and species level. When a complete taxonomical name was not possible to assign to species level, they are

referred as unknown (UNK) species. Bacterial species in which abundance differed statistically significantly (p<0.01) between uncomplicated and complicated acute appendicitis (S1 Table) and that were present at average relative abundance >0.1%, are illustrated in Fig 2C. Seven species that were more abundant in uncomplicated acute appendicitis were *Aggregatibacter aphrophilus*, *B. fragilis*, *Enterobacteriaceae UNK*, *Haemophilus UNK*, *Streptococcus UNK*, *Veillonella parvula*, and *Veillonella UNK*. Seven species were more abundant in complicated acute appendicitis: *Bacteroides faecis*, *Christensenellaceae UNK*, *Clostridia UCG-014*, *Dialister UNK*, *Oscillospiraceae UCG-005 UNK*, *Phocaeicola abscessus*, and *Porphyromonas endodontalis*. Genus *Aggregatibacter* was more abundant in uncomplicated acute appendicitis (adj. p = 0.00016), where the mean relative abundance was 6.4% compared with 1.0% in complicated acute appendicitis. Nine other, significantly different genera were *Veillonella* and an unknown genus from the family *Enterobacteriaceae*, which were enriched in uncomplicated acute appendicitis and an unknown genus from *Christensenellaceae*, *Defluviitaleaceae UCG-011 UNK*, *Family XIII UCG-001*, *Oscillibacter*, *Paludicola*, *Phocaeicola*, and *Subdoligranulum* were more abundant in complicated acute appendicitis (S1 Table). The abundances of bacterial phyla were not significantly different between uncomplicated and complicated acute appendicitis.

## Species poor appendiceal microbiomes are predominated by specific bacterial species

The overall microbial profiles of the appendiceal lumen are illustrated at species level in Fig 3 and in phylum and genus level in S1 Fig. Interindividual variation in both the composition and diversity was high in both forms of appendicitis severity. Further, in some samples only few predominant species were present at significant levels reflected in the low values of alpha diversity (Fig 3), while in other samples, hundreds of species were present and comprised an even microbial profile. The appendiceal microbiome was dominated by one bacterial species with a relative abundance of more than 50% in 17% (7/41) and 12% (9/77) of patients with uncomplicated and complicated acute appendicitis, respectively. *B. fragilis* and *Escherichia UNK* appeared as predominating species regardless of appendicitis severity. *Aggregatibacter aphrophilus* or *segnis*, and *Streptococcus UNK* (Fig 3) were found to dominate only in uncomplicated appendicitis and *Fusobacterium UNK*, *Haemophilus parainfluenzae*, *B. faecis*, *B. dorei* or *Bacteroides UNK* only in complicated appendicitis. Altogether 419 different genera and 599 different species were identified. Mean relative abundances of bacterial species composing more than 0.1% in each group are presented in S2 Table.

## Discussion

In this prospective clinical study on 118 patients with confirmed acute appendicitis, multiple differences between uncomplicated (n = 41) and complicated acute appendicitis (n = 77) patient samples were detected in the microbial composition of the appendiceal lumen supporting the theory of dividing acute appendicitis into two separate forms of acute inflammation processes with different clinical presentation and fates. There were differences in overall alpha and beta diversities between uncomplicated and complicated acute appendicitis. In addition, several bacterial taxa were identified to be significantly different between these two forms of appendicitis severity.

The microbial signatures in uncomplicated and complicated acute appendicitis were different, but in both forms of appendicitis severity, we found several opportunistic pathogens. Patients with uncomplicated acute appendicitis presented with a higher proportion of gram negative *Aggregatibacter species (spp)* consisting mainly of *A. segnis and A. aphrophilus (*former

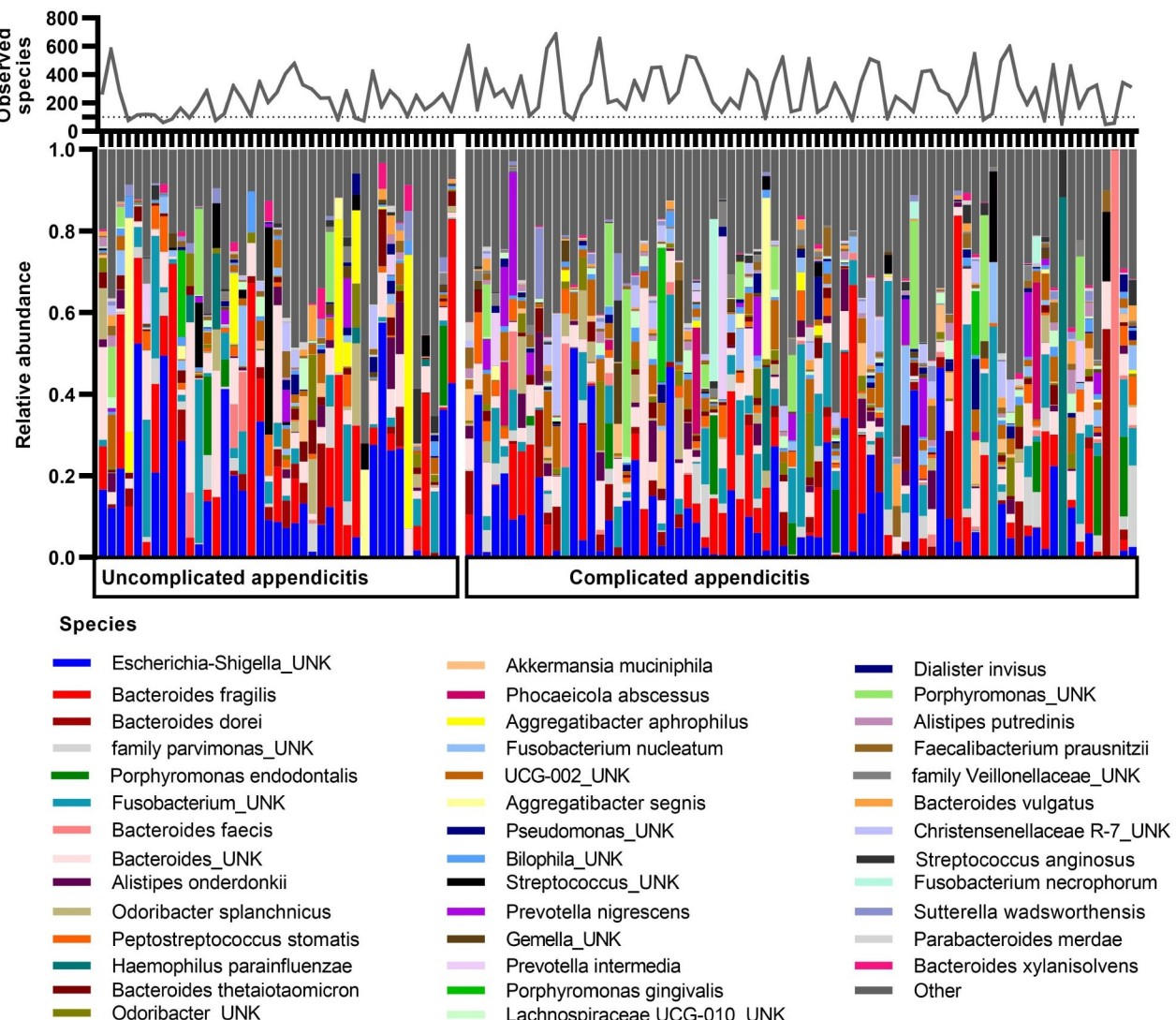

**Fig 3. Stacked barplots of appendiceal microbiome (n = 118) in species level in uncomplicated and complicated acute appendicitis.** Microbiome profiles of individual appendix samples with the 40 most abundant species plotted and lower abundance species grouped to "other". On top of barchart is overlaid the number of observed species in each sample as a line chart. UNK = unknown species.

*Haemophilus aphrophilus*). Aggregatibacter belongs to the HACEK group (gram negative bacteria *Haemophilus*, *Aggregatibacter*, *Cardiobacterium*, *Eikenella*, and *Kingella*) that are most notable for causing infective endocarditis, but they are also significant causes of periodontitis, abscesses, and septic arthritis [29]. In addition *Veillonella parvula* and an unknown species from the genus *Streptococcus* both linked to gastrointestinal tract and oral microbiota [30,31], were found to be enriched in uncomplicated acute appendicitis. In complicated acute appendicitis, a group of gram negative bacteria *Porphyromonas endodontalis*, an unknown species from the genus *Dialister*, and *Phocaeicola abscessus* were more abundant and all of these are considered to be oral pathogens [32,33]. Genera *Porphyromonas* and *Dialister* have been previously associated with complicated appendicitis in pediatric patients [18,34].

Microbial diversity was lower in uncomplicated acute appendicitis compared to complicated acute appendicitis. In both forms of appendicitis severity, we observed individual samples with very low diversity. These species poor appendiceal microbiomes were often

dominated by one bacteria representing at least 50% of the appendiceal microbiota. The observed predominance may play a role as an etiological factor on an individual level as it could be an indication of infection by these specific predominant species. Both uncomplicated and complicated acute appendicitis had microbial profiles where either *B. fragilis* or *Escherichia UNK* were the predominant bacteria, which are well reported in appendicitis [13,35]. *A. aphrophilus* or *A. segnis* and *Streptococcus UNK* were predominant only in patients with uncomplicated acute appendicitis. The most abundant *Fusobacteria* species *F. UNK* and *F. nucleatum* were present in both disease forms. However *F. UNK* as the predomint species was discovered only in the complicated appendicitis samples. *Fusobacteria* is the only group prominently associated with appendicitis [12,16] and our results corroborate the previous studies suggesting it to be specifically associated with complicated acute appendicitis [16].

Based on the results of this study, both infection and a disturbance of normal appendiceal microbiota could be etiological factors for some patients in both uncomplicated and complicated acute appendicitis. However, the observed difference in the microbiota diversity suggests that an infection may be more common in uncomplicated acute appendicitis partially associated with different bacterial species. This is further supported by the presence of an appendicolith in the majority (74%) of patients presenting with complicated acute appendicitis as appendicoliths often lead to obstruction of the appendiceal lumen [36]. In large RCTs, the presence of an appendicolith has been shown to be associated with a more complicated course of the disease [8,9].

This study has several limitations. First, a major limitation of the study is the lack of a healthy non-appendicitis control group. The perception of appendicitis as an opportunistic infection, where the abruption of the healthy appendiceal microbiota would drive the inflammation, is gaining ground [12,13]. In the search for etiological agents of a disease where the infection site contains normal microbiota, a healthy control group would be crucial. However, the acquisition of healthy appendix and appendiceal microbiota for this purpose is naturally both clinically and ethically impossible as the negative appendicectomy rate with the current CT diagnostic accuracy is very low and appendicectomy for healthy patients would be unethical. Second, further limitation is that patients received standard prophylactic antibiotics preoperatively, which could have affected the microbial composition of the appendiceal microbiota. However, all patients received antibiotics and the time between antibiotics administration and appendicectomy was relatively short. Third weakness of the study is the utilization of 16S rRNA gene amplicon sequencing, which is partly insufficient in species level analysis. Strain level identification is required in some bacteria to fully estimate the pathogenicity of bacteria present in the sample. For example, it has been suggested that specific virulent strains of *E. coli* [37] or the enterotoxic strains of *B. fragilis* [38] might be involved in the infection of the appendix and the resolution of taxonomical information gained in this study characterising appendiceal microbiota can be considered too low to drawn conclusions about the role of these bacteria in different forms of appendicitis severity. Future studies should harness methods, such as shotgun metagenomic sequencing combined with culturing methods, enabling the assessment of appendiceal microbiome function and virulence.

To our knowledge, this is the first prospective study with a large patient cohort and standardized clinical definitions of appendicitis severity comparing the microbial composition of the appendix in adult patients between uncomplicated and complicated acute appendicitis. The thorough prospective clinical data is a major strength in our study in addition to the so far largest number of patients.

In conclusion, uncomplicated and complicated acute appendicitis have different appendiceal microbiome profiles further supporting the disconnection between these two different forms of appendicitis severity.

## Supporting information

**S1 Fig. Appendiceal microbiome in phylum and genus level in uncomplicated and complicated acute appendicitis.**
(PDF)

**S1 Table. Differentially abundant bacterial species and genera between uncomplicated and complicated appendicitis.**
(PDF)

**S2 Table. Relative abundance of species in uncomplicated and complicated appendicitis.**
(PDF)

## Acknowledgments

We thank Aluke Oy, which contributed to the bioinformatic analysis of sequencing data. We thank our research coordinator Susanna Kulmala and laboratory technician Anna Musku. In addition we thank all of the surgeons on-call who took part in the enrollment of MAPPAC patients and collected the microbiological samples.

## Author Contributions

**Conceptualization:** Eveliina Munukka, Suvi Sippola, Juha Grönroos, Antti J. Hakanen, Paulina Salminen.

**Data curation:** Sanja Vanhatalo, Suvi Sippola, Jussi Haijanen, Paulina Salminen.

**Formal analysis:** Sanja Vanhatalo, Eveliina Munukka, Teemu Kallonen.

**Funding acquisition:** Sanja Vanhatalo, Eveliina Munukka, Juha Grönroos, Antti J. Hakanen, Paulina Salminen.

**Investigation:** Sanja Vanhatalo, Teemu Kallonen, Jussi Haijanen, Antti J. Hakanen, Paulina Salminen.

**Methodology:** Sanja Vanhatalo, Eveliina Munukka, Teemu Kallonen, Suvi Sippola, Juha Grönroos, Antti J. Hakanen, Paulina Salminen.

**Project administration:** Antti J. Hakanen, Paulina Salminen.

**Resources:** Juha Grönroos, Antti J. Hakanen, Paulina Salminen.

**Software:** Teemu Kallonen.

**Supervision:** Antti J. Hakanen, Paulina Salminen.

**Validation:** Eveliina Munukka, Teemu Kallonen, Paulina Salminen.

**Visualization:** Sanja Vanhatalo.

**Writing – original draft:** Sanja Vanhatalo.

**Writing – review & editing:** Sanja Vanhatalo, Eveliina Munukka, Teemu Kallonen, Suvi Sippola, Juha Grönroos, Jussi Haijanen, Antti J. Hakanen, Paulina Salminen.

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
