## [Decision Letter · Decision Letter 0]

26 Jul 2022

PONE-D-22-06832Appendiceal microbiome in uncomplicated and complicated acute appendicitis: a prospective cohort study (MAPPAC)PLOS ONE

Dear Dr. Salminen,

Thank you for submitting your manuscript to PLOS ONE. After careful consideration, we feel that it has merit but does not fully meet PLOS ONE’s publication criteria as it currently stands. Therefore, we invite you to submit a revised version of the manuscript that addresses the points raised during the review process.

I would like to sincerely apologise for the delay you have incurred with your submission. It has been exceptionally difficult to secure reviewers to evaluate your study. We have now received three completed reviews; the comments are available below. The reviewers have raised significant scientific concerns about the study, in particular about the study design and the lack of control group, that need to be addressed in a revision.

Please revise the manuscript to address all the reviewer's comments in a point-by-point response in order to ensure it is meeting the journal's publication criteria. Please note that the revised manuscript will need to undergo further review, we thus cannot at this point anticipate the outcome of the evaluation process.

We look forward to receiving your revised manuscript.

Kind regards,

Miquel Vall-llosera Camps

Senior Editor

PLOS ONE

3. You indicated that you had ethical approval for your study. In your Methods section, please ensure you have also stated whether you obtained consent from parents or guardians of the minors included in the study or whether the research ethics committee or IRB specifically waived the need for their consent.

4. Thank you for stating the following in the Funding Section of your manuscript:

“The MAPPAC study was supported by research grants from the Mary and Georg C. Ehrnrooth Foundation, the Sigrid Jusélius Foundation, the Finnish Academy, Government research grant awarded to Turku University Hospital (EVO foundation), The Maud Kuistila Memorial Foundation, Paulo Foundation, Doctoral Programme in Clinical Research of University of Turku, and Turku University foundation.”

“The MAPPAC study was supported by research grants from the Mary and Georg C. Ehrnrooth Foundation, the Sigrid Jusélius Foundation, the Finnish Academy, Government research grant awarded to Turku University Hospital (EVO foundation), The Maud Kuistila Memorial Foundation, Paulo Foundation,  Doctoral Program in Clinical Research at the University of Turku, and Turku University foundation. The funders had no role in study design, data collection and analysis, decision to publish, or preparation of the manuscript”

“EM is currently working as full-time Medical Advisor for Biocodex Nordics. PS reports receiving personal fees for lectures from Merck and Orion Pharma. AJH reports receiving personal fees for lectures from BioCodex, Merck and Pfizer. All other authors declare no competing interests.”

6. We note that you have indicated that data from this study are available upon request. PLOS only allows data to be available upon request if there are legal or ethical restrictions on sharing data publicly. For more information on unacceptable data access restrictions, please see http://journals.plos.org/plosone/s/data-availability#loc-unacceptable-data-access-restrictions.

Reviewers' comments:

Reviewer's Responses to Questions

**Comments to the Author**

1. Is the manuscript technically sound, and do the data support the conclusions?

Reviewer #1: Yes

Reviewer #2: Yes

Reviewer #3: Yes

2. Has the statistical analysis been performed appropriately and rigorously? 

Reviewer #1: Yes

Reviewer #2: Yes

Reviewer #3: Yes

3. Have the authors made all data underlying the findings in their manuscript fully available?

Reviewer #1: Yes

Reviewer #2: Yes

Reviewer #3: No

4. Is the manuscript presented in an intelligible fashion and written in standard English?

Reviewer #1: Yes

Reviewer #2: Yes

Reviewer #3: Yes

5. Review Comments to the Author

Reviewer #1: In this manuscript, the authors describe the examination of the appendiceal microbiome comparing patients with complicated and uncomplicated appendicitis. They appear to show differences in the microbiome as their main finding.

While the idea is great and important, there are significant issues that create confounders. One of the most important issues is that there is no control group. I know that the authors recognize this in their discussion as a limitation. However, this is a huge limitation for a number of reasons outlined below.

In table 1, the authors outline the types of complications. Many of these are concerning in that they might create irregularities in data. For example, an appendicolith may simply be a stool fragment imbedded in the appendix. During sequencing would the microbiome of the appendicolith be what is determined rather than the appendicitis?

Another concern is the perforation. In essences the perforation means the appendix is "draining"...albeit into the peritoneal cavity. Nonetheless it is decompressing the appendix. There are a number of events that might happen in this state. First the pathogenic microbes may diminish. Secondly, the perforated appendix patient may be more symptomatic with systemic symptoms such as fever. As a result is more likely to be on antibiotics. What antibiotics may or may not have been given to subjects? I cannot find a list of antibiotics or the use of antibiotics. This will most certainly influence the microbiome results.

To correct these, at a minimum, the authors should disclose all antibiotic use and to which group these were used. Secondly, the number of days of antibiotics. Also, the number of days that the patients had symptoms and whether fever or other factors were present. Also, to analyzed perforation with out abscesss, appendicolith or tumor, would be helpful.

Reviewer #2: Important note: This review pertains only to ‘statistical aspects’ of the study and so ‘clinical aspects’ [like medical importance, relevance of the study, ‘clinical significance and implication(s)’ of the whole study, etc.] are to be evaluated [should be assessed] separately/independently. Further please note that any ‘statistical review’ is generally done under the assumption that (such) study specific methodological [as well as execution] issues are perfectly taken care of by the investigator(s). This review is not an exception to that and so does not cover clinical aspects {however, seldom comments are made only if those issues are intimately / scientifically related & intermingle with ‘statistical aspects’ of the study}. Agreed that ‘statistical methods’ are used as just tools here, however, they are vital part of methodology [and so should be given due importance].

COMMENTS: In my opinion, since this study was a single-center arm of a multicenter MAPPAC (Microbiology Appendicitis Acuta) trial, the study should be treated as having different ‘Study design and methodology’. Therefore, quoting previously published protocol describing these [Study design and key methods have been reported previously (30)] is not correct. This quoted ‘study protocol is for the ‘MAPPAC trial’ and is definitely excellent, however, since this publication is not ‘one of the series’, giving sufficient details regarding ‘Study design and methodology’ are expected. That protocol is for two arms two clinical trials but the present one is single arm study. How can the ‘Study design and methodology’ be same? Therefore, is that really relevant quoting previously published protocol? In ‘Results’ section saying that “Altogether 308 patients were enrolled in the MAPPAC study between April 11, 2017, and March 29, 2019; Figure 1 shows the study flow” do you think has any meaning/relevance?

May I request you to make a small change in title of study [“Appendiceal microbiome in uncomplicated and complicated acute appendicitis: a prospective cohort study (MAPPAC)”] because ‘MAPPAC’ is another trial. This is only a suggestion (as inclusion of the term ‘MAPPAC’) may cause confusion and average reader like me, will take the present study as MAPPAC. You may not follow this as this one is my subjective opinion [please free to take appropriate decision].

Since this study is based on ‘single-arm design’, may I further request to read the following [which is pasted from one standard textbook on ‘Research Methodology’]:

For a pilot study it is alright to have ‘single-arm design’, or it is alright when that is the only possibility’, however, It is very essential to keep the limitations in mind while interpreting results. Further, note that a classical/ideal clinical trial/study needs/requires a concurrently {but similarly} handled/treated appropriately selected/chosen control/comparison parallel group/arm.

Note further that “Inferential statistics (i.e., hypothesis testing + estimation of CI) is built on the population model [which means the underlying assumption is that there is/are population(s) and we are dealing with random sample(s) drawn from that/those population(s)]. Although in clinical trial (involving at least two groups) we do not really deal with random samples (generally a non-probabilistic convenience sampling), ‘allocation’ to treatment groups is ‘randomly’ done which enable us to evoke the population model and we can use inferential statistics safely. But when there is only one group (so that there is no question of random allocation), with ‘non-random’ selection, it may be questionable to use inferential statistics even if you have two measurement sets as ‘pre-post’ or use ‘internal grouping for comparison” [like here uncomplicated acute appendicitis compared to complicated appendicitis but this is an ‘internal grouping’].

I am sure that these learned authors already know these things, however, it is very essential to keep the limitations in mind while interpreting results {note that I am not asking you to change the study design}.

What exactly do you mean by ‘excluded due to failed library preparation (n=26)’? Please note that any ‘regression models / regression techniques’ are not originally developed for head-to-head between group comparison(s), [Alpha diversity analyses were performed using standard linear regression models with alpha diversity measure (Chao1, Shannon entropy or number of observed species) as the response variable]. Nevertheless, limitations of the study highlighted on page 14 are highly appreciated. However, as pointed out in ‘important note’ above “This review pertains only to ‘statistical aspects’ of the study and so ‘clinical aspects’ should be assessed separately/independently [one should carefully consider/look at the clinical implications of the study].

One last minor doubt, refer to page 15 where you said “To our knowledge, this is the first prospective study with a large patient cohort and standardized clinical definitions of appendicitis severity comparing the microbial composition of the appendix (may be true) in adult patients between uncomplicated and complicated acute appendicitis” and quoted reference number 14 for this. My doubt is ‘What evidence this reference provided?’ to say you so or ‘what in this reference is indicative of this fact?’.

Except these minor points, in my opinion, this article/manuscript is good and needs a small amount of re-vision (which is quite possible). After minor revision, this manuscript may be accepted [provided clinical implications of these findings/results are valuable].

Reviewer #3: I have read a very interesting manuscript and I provide the following minor queries for the authors to address.

1. Provide a succinct definition of how was a complicated versus a non complicated acute appendicitis defined.

2. Microbiome dysbiosis of the appendix has been recently advanced as causal re the pathogenesis of appendicits, was there scope from the data reported to further highlight appendix microbiome dysbiosis as a feature in this study?

3. The presence of oral bacterial pathogens in the large bowel and consequently the appendix elicits the query as to how do these pathogens travel to these sites. As such then were any patients on proton pump inhibitors that in reducing stomach acid reduces this barrier to orally ingested microbes? This perhaps explaining in part how a pathogen such as Porphyromonas endodontalis is in the appendix.

6. PLOS authors have the option to publish the peer review history of their article (what does this mean?). If published, this will include your full peer review and any attached files.

Reviewer #1: No

Reviewer #2: No

Reviewer #3: **Yes: **Luis Vitetta

---

## [Author Response · Author response to Decision Letter 0]

24 Aug 2022

Response to Reviewers 

PONE-D-22-06832

Appendiceal microbiome in uncomplicated and complicated acute appendicitis: a prospective cohort study (MAPPAC)

PLOS ONE

https://journals.plos.org/plosone/s/file?id=wjVg/PLOSOne_formatting_sample_main_body.pdfand

Our response: We have revised the manuscript according to PLOS ONE’s style requirements and named the files accordingly.

2. Please update your submission to use the PLOS LaTeX template. The template and more information on our requirements for LaTeX submissions can be found athttp://journals.plos.org/plosone/s/latex.

Our response: We apologize for failing to complete this requirement as using LaTeX is not possible at our institution. If needed, the .doc-file hopefully and likely can be converted to a LateX file during the editorial process, if our manuscript is accepted for publication.

3. You indicated that you had ethical approval for your study. In your Methods section, please ensure you have also stated whether you obtained consent from parents or guardians of the minors included in the study or whether the research ethics committee or IRB specifically waived the need for their consent.

Our response: This information has now been added: “Trial was performed in accordance with the Declaration of Helsinki and all patients or in the case of minors, their legal guardians gave written informed consent to participate in the study.”

4. Thank you for stating the following in the Funding Section of your manuscript:

“The MAPPAC study was supported by research grants from the Mary and Georg C. Ehrnrooth Foundation, the Sigrid Jusélius Foundation, the Finnish Academy, Government research grant awarded to Turku University Hospital (EVO foundation), The Maud Kuistila Memorial Foundation, Paulo Foundation, Doctoral Programme in Clinical Research of University of Turku, and Turku University foundation.”

“The MAPPAC study was supported by research grants from the Mary and Georg C. Ehrnrooth Foundation, the Sigrid Jusélius Foundation, the Finnish Academy, Government research grant awarded to Turku University Hospital (EVO foundation), The Maud Kuistila Memorial Foundation, Paulo Foundation, Doctoral Program in Clinical Research at the University of Turku, and Turku University foundation. The funders had no role in study design, data collection and analysis, decision to publish, or preparation of the manuscript”

Our response: The current funding statement is correct; we have added that in the cover letter and omitted all funding information from the manuscript. Thank you for revising this for us in the online system.

“EM is currently working as full-time Medical Advisor for Biocodex Nordics. PS reports receiving personal fees for lectures from Merck and Orion Pharma. AJH reports receiving personal fees for lectures from BioCodex, Merck and Pfizer. All other authors declare no competing interests.”

Our response: We apologize for missing this requirement in the original submission and we have now added the required statement in the competing interests section, which is added to the cover letter. 

6. We note that you have indicated that data from this study are available upon request. PLOS only allows data to be available upon request if there are legal or ethical restrictions on sharing data publicly. For more information on unacceptable data access restrictions, please see http://journals.plos.org/plosone/s/data-availability#loc-unacceptable-data-access-restrictions.

Our response: We have revised the data availability statement accordingly. 

a) The clinical patient data is available only by request to the primary investigator and this data includes sensitive information despite the de-identification and thus cannot be shared based on legal restrictions by Finnish GDPR rules. The point of contact is the primary investigator and this revised statement is provided in the cover letter. 

b) The raw 16S-data can be shared and the anonymized data set is now uploaded in a public repository: http://www.ncbi.nlm.nih.gov/bioproject/869932 . Dataset can be previewed before publication at https://dataview.ncbi.nlm.nih.gov/object/PRJNA869932?reviewer=prdj5aj31och08c57fdv24devt

Reviewer comments

Reviewer #1

1. In this manuscript, the authors describe the examination of the appendiceal microbiome comparing patients with complicated and uncomplicated appendicitis. They appear to show differences in the microbiome as their main finding.

While the idea is great and important, there are significant issues that create confounders. One of the most important issues is that there is no control group. I know that the authors recognize this in their discussion as a limitation. However, this is a huge limitation for a number of reasons outlined below. 

Our response: We thank the reviewer for constructive comments and fully agree that the lack of a healthy control group is a clear limitation as we have stated in the study limitations. However, as stated in the manuscript, it would be both clinically and ethically impossible to have a control group as there is no clinical indication of appendectomy for an uninflamed appendix. Removal of a healthy appendix is associated with similar complication rate as appendectomy for acute appendicitis and furthermore, the role of the appendix in general is still unclear underlining the impossibility of subjecting patients to appendectomy to achieve a healthy control group. Negative appendicectomy rate is currently very low due to high accuracy of the preintervention CT diagnostics, i.e. these few cases would not constitute such a healthy control group that would be needed. These issues have already been stated in the study limitations as follows and no further additions have been made: “However, the acquisition of healthy appendix and appendiceal microbiota for this purpose is naturally both clinically and ethically impossible as the negative appendicectomy rate with the current CT diagnostic accuracy is very low and appendicectomy for healthy patients would be unethical.”

2. In table 1, the authors outline the types of complications. Many of these are concerning in that they might create irregularities in data. For example, an appendicolith may simply be a stool fragment imbedded in the appendix. During sequencing would the microbiome of the appendicolith be what is determined rather than the appendicitis?

Our response: We thank you the reviewer for pointing out this important fact. Most recognized definition of the complicated appendicitis is the presence of gangrene, abscess, or perforation. Further, the presence of appendicolith has been shown to be associated with a more complicated form of the disease. However, in this patient cohort and in all concurrent trials, an appendicolith was a clearly visible concretion on CT and not a merely a stool fragment in the appendix. The appendicoliths can easily be avoided and have been avoided during the sample collection, i.e. our analysis actually implies the appendiceal microbiome and not the microbiome of an appendicolith. We do recognize that these cannot be fully isolated from each other and that the presence of and appendicolith may very well have an effect on the appendiceal microbiome. 

3. Another concern is the perforation. In essences the perforation means the appendix is "draining"...albeit into the peritoneal cavity. Nonetheless it is decompressing the appendix. There are a number of events that might happen in this state. First the pathogenic microbes may diminish. Secondly, the perforated appendix patient may be more symptomatic with systemic symptoms such as fever. As a result is more likely to be on antibiotics. What antibiotics may or may not have been given to subjects? I cannot find a list of antibiotics or the use of antibiotics. This will most certainly influence the microbiome results.

Our response: We agree with the reviewer that perforated appendicitis does exactly decompress the appendix and as it is complicated acute appendicitis, it may be associated with more systemic symptoms as stated by the reviewer. Please see our response to comment #4 for the antibiotic use. In addition, the number of patients in the subgroups of complicated acute appendicitis are very small limiting the ability of any subgroup analysis in providing comparative outcomes data between these different groups. Thus these groups are only described in table 1 as no conclusions can be drawn in comparing these groups as pointed out by reviewer #2, comment #3.

4. To correct these, at a minimum, the authors should disclose all antibiotic use and to which group these were used. Secondly, the number of days of antibiotics. Also, the number of days that the patients had symptoms and whether fever or other factors were present. Also, to analyzed perforation with out abscesss, appendicolith or tumor, would be helpful.

Our response: We agree with the reviewer that the antibiotic usage is a very relevant point and thank the reviewer for pointing out this. For acute appendicitis, there is standard practice of antibiotic prophylaxis 30 minutes prior to appendectomy and this is valid for all appendicitis cases, i.e. none of the patients, not even the complicated cases, have a longer preoperative antibiotic duration. The effect of mere antibiotic prophylaxis given 30 minutes preoperatively presumably has either little or no effect to the appendiceal microbiota. As appendectomy is performed within the first 6-8 hours especially for the complicated acute appendicitis cases, even these patients did not have any major effect of antibiotics on the appendiceal microbiota. We have now clarified this information in the Methods section as follows:

“Study design and patients: The study enrolled adult patients having either uncomplicated or complicated acute appendicitis confirmed either CT or clinically. Patients are undergoing appendicectomy in order to have the appendix both as the reference standard for the clinical diagnosis and the availability of appendiceal samples. According to standard clinical practice, patients received antibiotic prophylaxis as a single dose of 1.5 g cefuroxime and 500 mg metronidazole approximately 30 minutes prior to appendectomy initiation. None of the patients had long antibiotic treatment durations preoperatively.”

In addition, the sequencing method does not actually differentiate between dead and living bacteria. 

The duration of symptoms from a clinical point may not be that relevant as we have the information on appendicitis severity from a combination of CT, surgical finding, and histopathology. Fever is an important clinical parameter, but most likely with the differential diagnosis based on the previously mentioned findings, analyzing patient temperatures most likely may not add to the current analysis. In addition, the patients may have taken or received antipyretic medications further adding potential bias in adding this information. However, if this is evaluated necessary by the editors, patient temperature on admission can naturally be reported for both groups. 

Regarding the subgroup analysis of the patients with perforation only, we agree with the reviewer that it would be interesting to assess the potential differences of the various forms of complicated acute appendicitis. For this, the number of patients is too low to draw any conclusions (n=14) and we have thus not performed this subgroup analysis

Reviewer #2 

Important note: This review pertains only to ‘statistical aspects’ of the study and so ‘clinical aspects’ [like medical importance, relevance of the study, ‘clinical significance and implication(s)’ of the whole study, etc.] are to be evaluated [should be assessed] separately/independently. Further please note that any ‘statistical review’ is generally done under the assumption that (such) study specific methodological [as well as execution] issues are perfectly taken care of by the investigator(s). This review is not an exception to that and so does not cover clinical aspects {however, seldom comments are made only if those issues are intimately / scientifically related & intermingle with ‘statistical aspects’ of the study}. Agreed that ‘statistical methods’ are used as just tools here, however, they are vital part of methodology [and so should be given due importance].

COMMENTS: 

1. In my opinion, since this study was a single-center arm of a multicenter MAPPAC (Microbiology Appendicitis Acuta) trial, the study should be treated as having different ‘Study design and methodology’. Therefore, quoting previously published protocol describing these [Study design and key methods have been reported previously (30)] is not correct. This quoted ‘study protocol is for the ‘MAPPAC trial’ and is definitely excellent, however, since this publication is not ‘one of the series’, giving sufficient details regarding ‘Study design and methodology’ are expected. That protocol is for two arms two clinical trials but the present one is single arm study. How can the ‘Study design and methodology’ be same? Therefore, is that really relevant quoting previously published protocol? In ‘Results’ section saying that “Altogether 308 patients were enrolled in the MAPPAC study between April 11, 2017, and March 29, 2019; Figure 1 shows the study flow” do you think has any meaning/relevance?

Our response: We thank the reviewer for the thorough feedback and understand the issues pointed out by the reviewer. However, from a clinical perspective it is of vital importance to describe the recruitment of the patients in the larger MAPPAC trial in concurrence with the APPAC II and APPAC III trials as this is the basis for describing the whole patient population to the readers adding to the understanding of how representative this single-center study arm patient population is of the larger patient cohort. We have already stated the single-center arm nature of this study in the methods and we have further clarified the study rationale in the Methods section as follows.

”This prospective cohort study was a single-center arm of a multicenter MAPPAC (Microbiology Appendicitis Acuta) trial (ClinicalTrials.gov NCT03257423) and was conducted at Turku University Hospital in Finland. Study rationale and key methods have been reported previously (20).”

2. May I request you to make a small change in title of study [“Appendiceal microbiome in uncomplicated and complicated acute appendicitis: a prospective cohort study (MAPPAC)”] because ‘MAPPAC’ is another trial. This is only a suggestion (as inclusion of the term ‘MAPPAC’) may cause confusion and average reader like me, will take the present study as MAPPAC. You may not follow this as this one is my subjective opinion [please free to take appropriate decision].

Our response: Thank you for the suggestion. Please see our response to comment #1, this study is a key part of the MAPPAC trial, but we do understand the reviewer’s point of view and we have omitted the trial name from the title. 

3. Since this study is based on ‘single-arm design’, may I further request to read the following [which is pasted from one standard textbook on ‘Research Methodology’]:

For a pilot study it is alright to have ‘single-arm design’, or it is alright when that is the only possibility’, however, It is very essential to keep the limitations in mind while interpreting results. Further, note that a classical/ideal clinical trial/study needs/requires a concurrently {but similarly} handled/treated appropriately selected/chosen control/comparison parallel group/arm.

Note further that “Inferential statistics (i.e., hypothesis testing + estimation of CI) is built on the population model [which means the underlying assumption is that there is/are population(s) and we are dealing with random sample(s) drawn from that/those population(s)]. Although in clinical trial (involving at least two groups) we do not really deal with random samples (generally a non-probabilistic convenience sampling), ‘allocation’ to treatment groups is ‘randomly’ done which enable us to evoke the population model and we can use inferential statistics safely. But when there is only one group (so that there is no question of random allocation), with ‘non-random’ selection, it may be questionable to use inferential statistics even if you have two measurement sets as ‘pre-post’ or use ‘internal grouping for comparison” [like here uncomplicated acute appendicitis compared to complicated appendicitis but this is an ‘internal grouping’].

I am sure that these learned authors already know these things, however, it is very essential to keep the limitations in mind while interpreting results {note that I am not asking you to change the study design}.

Our response: We thank the reviewer for the constructive criticism and the MAPPAC study in itself is not a randomized controlled trial, but the patients recruited to the MAPPAC trial were partially patients enrolled in two concurrent RCTs and partially patients excluded from these trials. This current single-center study is a prospective cohort study, as stated in the title, comparing the appendiceal microbiota in uncomplicated and complicated acute appendicitis in patients excluded from the RCTs as all of these patients in this single-center study underwent appendectomy. Based on the study design, we fully agree with the reviewer that this limitation has to be kept in mind when interpreting the results as the results are more descriptive and comparison between the two groups is more limited in this study design. We have thus aimed to focus on only describing the microbial findings i.e. bacterial profiles and signatures within the samples and further report potential associations between appendicitis severity and appendiceal microbiota. Since uncomplicated and complicated appendicitis are recognized as two different forms of acute appendicitis, the comparisons are essentially made between two populations with representative samples rather than internal grouping comparison.

We have now further underlined this limitation by revising the conclusion as follows. Abstract and Discussion conclusion: “Uncomplicated and complicated acute appendicitis seem to have different appendiceal microbiome profiles further supporting the disconnection between these two different forms of acute appendicitis.”

4. What exactly do you mean by ‘excluded due to failed library preparation (n=26)’? Please note that any ‘regression models / regression techniques’ are not originally developed for head-to-head between group comparison(s), [Alpha diversity analyses were performed using standard linear regression models with alpha diversity measure (Chao1, Shannon entropy or number of observed species) as the response variable]. Nevertheless, limitations of the study highlighted on page 14 are highly appreciated. However, as pointed out in ‘important note’ above “This review pertains only to ‘statistical aspects’ of the study and so ‘clinical aspects’ should be assessed separately/independently [one should carefully consider/look at the clinical implications of the study].

Our response: In Results section, we report that 26 samples originally collected were not included in final statistical analyses. The PCR amplification step in 16S rRNA amplicon library preparation protocol of these 26 samples failed to produce enough DNA in order to proceed to sequencing. Criteria for a successful library preparation has now been added to methods section.

”16S rRNA amplicon sequencing

16S rRNA amplicon sequencing was performed targeting the V3-V4 hypervariable region. Negative and positive control samples were included in the sequencing: negative DNA extraction control, negative PCR control, and a mock community (ZymoBiomics microbial community DNA standard, Zymo Research, Irvine, California, USA) as a positive control. Amplicon libraries were generated following the Illumina protocol (https://support.illumina.com/documents/documentation/chemistry_documentation/16s/16s-metagenomic-library-prep-guide-15044223-b.pdf) with the exception that increased amount of 75 ng of DNA template was used in the amplicon PCR reaction. Amplicon PCR products were verified (size, integrity) with DNA agarose gel and library preparation was considered successful if a clear and right sized (550 bp) DNA band was seen on gel. Amplicon libraries were quantified using Qubit fluorometer and 10% of Phix (Illumina, San Diego, California, USA) was added to each equimolar pool. Sequencing was performed using MiSeq Reagent Kit v3 and paired-end 2×300 bp protocol on a MiSeq System (Illumina).”

5. One last minor doubt, refer to page 15 where you said “To our knowledge, this is the first prospective study with a large patient cohort and standardized clinical definitions of appendicitis severity comparing the microbial composition of the appendix (may be true) in adult patients between uncomplicated and complicated acute appendicitis” and quoted reference number 14 for this. My doubt is ‘What evidence this reference provided?’ to say you so or ‘what in this reference is indicative of this fact?’.

Our response: Thank you for pointing out this mistake as the reference should not be there. We have now removed the reference from the sentence.

Except these minor points, in my opinion, this article/manuscript is good and needs a small amount of re-vision (which is quite possible). After minor revision, this manuscript may be accepted [provided clinical implications of these findings/results are valuable].

Reviewer #3

 I have read a very interesting manuscript and I provide the following minor queries for the authors to address.

1. Provide a succinct definition of how was a complicated versus a non complicated acute appendicitis defined.

Our response: We thank the reviewer for the suggestions and have now added the more accurate definition to manuscript:

” The CT criteria for uncomplicated acute appendicitis included appendiceal diameter exceeding 6 mm with thickened and enhanced appendiceal wall and periappendiceal edema and /or minor fluid collection. Intramural neutrophil invasion in the histopathological examination of the removed appendix was required for the diagnosis of acute appendicitis. Acute appendicitis was defined uncomplicated if no features of complicated acute appenditis was present. Complicated appendicitis was defined as the presence of an appendicolith, perforation, abscess, gangrene, or suspicion of tumor or the combination of these.”

2. Microbiome dysbiosis of the appendix has been recently advanced as causal re the pathogenesis of appendicits, was there scope from the data reported to further highlight appendix microbiome dysbiosis as a feature in this study?

Our response: Thank you for disclosing the interesting concept of microbiome dysbiosis. However, the definition of the dysbiosis in this study field is still very much in progress, and this is the case especially related to appendicitis. Our primary aim was simply to describe the microbial ecosystems within the appendix and evaluate the possible differences in the signatures and profiles between the two forms of appendicitis severity. For the future trials, we agree with the reviewer that microbiome dysbiosis of the appendix needs to be addressed after bridging these existing knowledge gaps of the appendiceal microbiome. 

3. The presence of oral bacterial pathogens in the large bowel and consequently the appendix elicits the query as to how do these pathogens travel to these sites. As such then were any patients on proton pump inhibitors that in reducing stomach acid reduces this barrier to orally ingested microbes? This perhaps explaining in part how a pathogen such as Porphyromonas endodontalis is in the appendix.

Our response: This is an interesting consideration pointed out by the reviewer. We do agree that the oral microbiome could have an effect on the appendiceal microbiome and this effect could be further enhanced by proton pump inhibitors. However, to investigate the role of specific oral pathogens such as Porphyromonas endodontalis in appendicitis etiology to our opinion it would require more profound analysis approach, such as shotgun metagenomic sequencing, in order to reach strain level identification of possible pathogens. The presence of oral bacterial pathogens and the medications potentially affecting the orally ingested microbes needs to be included in the scope of future research.

---

## [Decision Letter · Decision Letter 1]

27 Sep 2022

Appendiceal microbiome in uncomplicated and complicated acute appendicitis: a prospective cohort study

PONE-D-22-06832R1

Dear Dr. Paulina Salminen,

We’re pleased to inform you that your manuscript has been judged scientifically suitable for publication and will be formally accepted for publication once it meets all outstanding technical requirements.

Kind regards,

Vipa Thanachartwet, M.D.

Academic Editor

PLOS ONE

Additional Editor Comments (optional):

All issues raised by the reviewers are addressed.

Reviewers' comments:

Reviewer's Responses to Questions

**Comments to the Author**

1. If the authors have adequately addressed your comments raised in a previous round of review and you feel that this manuscript is now acceptable for publication, you may indicate that here to bypass the “Comments to the Author” section, enter your conflict of interest statement in the “Confidential to Editor” section, and submit your "Accept" recommendation.

Reviewer #2: (No Response)

Reviewer #3: All comments have been addressed

2. Is the manuscript technically sound, and do the data support the conclusions?

Reviewer #2: (No Response)

Reviewer #3: Yes

3. Has the statistical analysis been performed appropriately and rigorously? 

Reviewer #2: (No Response)

Reviewer #3: Yes

4. Have the authors made all data underlying the findings in their manuscript fully available?

Reviewer #2: (No Response)

Reviewer #3: Yes

5. Is the manuscript presented in an intelligible fashion and written in standard English?

Reviewer #2: (No Response)

Reviewer #3: (No Response)

6. Review Comments to the Author

Reviewer #2: COMMENTS: Since most of the comments made on earlier draft [though not all were/are attended/followed and I am not very much convinced for reasons given or arguments made (for example, in response to one query/comment you said “Since uncomplicated and complicated appendicitis are recognized as two different forms of acute appendicitis, the comparisons are essentially made between two populations with representative samples rather than internal grouping comparison”) with which I do not agree at all. I never said (and I can very-well understand it) that uncomplicated and complicated appendicitis are recognized as two different forms of acute appendicitis but how can the comparison was between two populations with representative samples when samples were not selected from these two populations?) and so definitely not happy], I recommend the acceptance, because the manuscript now has achieved acceptable level [I had earlier said that after minor revision (suggested), this manuscript may be accepted].

Reviewer #3: I have read the revised manuscript with all queries that have been addressed and as such I have no further remakrs to add.

7. PLOS authors have the option to publish the peer review history of their article (what does this mean?). If published, this will include your full peer review and any attached files.

Reviewer #2: **Yes: **Dr. Sanjeev Sarmukaddam

Reviewer #3: **Yes: **Luis Vitetta

---

## [Editor Report · Acceptance letter]

6 Oct 2022

PONE-D-22-06832R1 

Appendiceal microbiome in uncomplicated and complicated acute appendicitis: a prospective cohort study 

Dear Dr. Salminen:

I'm pleased to inform you that your manuscript has been deemed suitable for publication in PLOS ONE. Congratulations! Your manuscript is now with our production department. 

Kind regards, 

on behalf of

Associate Professor Vipa Thanachartwet 

Academic Editor

PLOS ONE